# Can Adjuvant HPV Vaccination Be Helpful in the Prevention of Persistent/Recurrent Cervical Dysplasia after Surgical Treatment?—A Literature Review

**DOI:** 10.3390/cancers14184352

**Published:** 2022-09-07

**Authors:** Kaja Michalczyk, Marcin Misiek, Anita Chudecka-Głaz

**Affiliations:** 1Department of Gynecological Surgery and Gynecological Oncology of Adults and Adolescents, Pomeranian Medical University, 70-204 Szczecin, Poland; 2Holy Cross Cancer Center, Clinical Gynecology, 25-743 Kielce, Poland

**Keywords:** HPV, vaccination, cervical dysplasia, CIN, HSIL, conization, LEEP

## Abstract

**Simple Summary:**

Primary prophylactic, early detection and the treatment of precancerous lesions are the main goals of cervical cancer screening. Despite effective surgical treatment methods, using loop electrosurgical excision procedures and conization, the overall risk of the recurrence of HSIL lesions remains at approximately 6.6%. There is increasing evidence of the potential role of HPV vaccines in the adjuvant setting and their impact on the reduction of disease recurrence. This review aims to analyze the up-to-date research concerning the use and efficacy of secondary human papilloma virus (HPV) vaccination as an adjuvant method to surgical treatment in patients diagnosed with cervical HSILs.

**Abstract:**

Cervical cancer formation is preceded by precursor lesions, including low-grade squamous intraepithelial lesions (LSILs) and high-grade squamous intraepithelial lesions (HSILs), which are usually diagnosed in women of reproductive age. Despite the recent advanced diagnostic and treatment methods, including colposcopy, the loop electrosurgical excision procedure (LEEP), and surgical conization, the recurrence or residual disease affects as many as 6.6% of patients. The lesions are often associated with human papilloma virus (HPV) infection. As HPV persistence is the leading and only modifiable factor affecting the risk of progression of CIN lesions into high-grade cervical dysplasia and cancer, it has been proposed to conduct adjuvant vaccination in patients treated for high-grade cervical dysplasia. To date, no vaccine has been approved for therapeutic use in patients diagnosed with HSILs; however, attempts have been made to determine the use of HPV prophylactic vaccination to reduce recurrent HSILs and prevent cervical cancer. The aim of this review was to analyze the up-to-date literature concerning the possible use of secondary human papilloma virus (HPV) vaccination as an adjuvant method to surgical treatment in patients diagnosed with cervical HSILs. Adjuvant HPV vaccination after surgical treatment may reduce the risk of recurrent cervical dysplasia.

## 1. Introduction

Each year, more than half a million women are diagnosed with cervical cancer [1]. Despite the advances in cancer screening and prevention, it remains one of the leading causes of cancer deaths. Chronic human papilloma virus (HPV) infection is responsible for 99.7% of cervical cancer diagnoses [2]. The virus is sexually transmitted; however, transmission does not require penetrative sex, as it can also occur via skin-to-skin contact, including anal and oral sex, and contact with genital warts [3]. HPV infection risk factors include early sexual debut, multiple sexual partners, immunosuppression, and a history of sexually transmitted infections. In accordance with Globocan [4], it is believed that over 70% of the sexually active population will be infected with HPV at some point in their lives. As of 2018, the World Health Organization (WHO) has called for action toward eliminating cervical cancer as a public health problem. A triple-intervention strategy has been established with specifically determined global targets to be reached by 2030: 90% of girls should be fully vaccinated with the HPV vaccine by the age of 15; 70% of women need to be screened with a high-performance test two times by 35 and 45 years of age; and 90% of women with the identified cervical disease need to receive adequate treatment and care [5]. Primary preventions using vaccinations, screening with the identification of high-risk HPV strains, and pap smears are the most effective methods to decrease cervical cancer burden and mortality. However, is there secondary prevention for patients already infected with HPV and diagnosed with cervical intraepithelial neoplasia?

Cervical cancer is preceded by precursor lesions, including low-grade squamous intraepithelial lesions (LSILs) and high-grade squamous intraepithelial lesions (HSILs), which are usually diagnosed in women of reproductive age. Despite the recent advanced diagnostic and treatment methods, including colposcopy, the loop electrosurgical excision procedure (LEEP), and surgical conization, the recurrence or residual disease affects as many as 6.6% of patients [6]. There are significant discrepancies in cervical cancer incidence and mortality based on geographic localization. Due to the introduction of formalized screening programs, its incidence has already halved, with the greatest decrease in cervical cancer prevalence occurring in high-income countries. However, in Africa and Latin America, cervical cancer remains the leading cause of female cancer mortalities [7].

HPV is a family of more than 200 types of non-enveloped double-stranded DNA viruses that can infect mucosal and cutaneous epithelial cells. They can be divided into high risk and low risk based on their oncogenic capacity. The most common high-risk viruses responsible for approximately 70% of persistent HPV infections are HPV types 16 and 18, while types 31, 33, 45, 52, and 58 account for 19% [6,8]. HPV is not only responsible for cervical cancer but also oropharyngeal, vulvar, vaginal, anal, and penile cancers [9]. Most HPV-induced changes are transient; as many as 90% will regress spontaneously within the 12 to 36 months following infection [10,11]. However, the persistence of HPV infection plays an important role in developing invasive cervical cancer due to the accumulation of mutations in the transformation zone it causes. Factors such as an individual’s genetic predisposition, including p53 polymorphism, genetic variation within the HPV type, the coinfection of multiple HPV types, the frequency of reinfection, hormone levels, and the patient’s immune response, may influence an individual’s infection course and clearance [2].

## 2. HPV and Oncogenesis

The HPV genome contains early (E) genes, which control DNA maintenance, replication, and transcription, while the late (L) genes encode proteins that build up the viral capsid. In the early stages of HPV infection, proteins E1 and E2 are expressed in significant amounts and allow viral replication within the infected tissues. In cervical HPV infection, the changes lead to the induction of low-grade squamous intraepithelial lesions. The best known mechanism responsible for malignant transformation involves viral oncoproteins E6 and E7, as they bind to the p53 tumor suppressor protein and retinoblastoma tumor suppressor protein (pRb), leading to the degradation of the suppressor proteins and allowing for cell proliferation and the initiation of carcinogenesis [12]. The integration of the viral genome and the dysregulation of the E2 protein, which regulates oncoproteins E6 and E7, result in their overexpression and uncontrolled cell growth. The virus infects cells in the basal layer. It carries out an infection cycle that is very similar to the differentiation program of the surrounding cells, making it difficult for the host’s immune system to recognize it [13].

HPV infection occurs via a microinjury from an infection or sexual contact when antigen-presenting cells (APCs) are exposed to the HPV virus. Monocyte–macrophage and dendritic cells localized in the epithelium induce immune responses by releasing proinflammatory cytokines, including interleukins IL-1, IL-6, and IL-12, and tumor necrosis factor-alpha (TNF-α) [14]. Host innate immune responses are crucial for early infection clearance, while later adaptive immune responses are necessary for the regression of already established lesions [15]. HPV has created multiple mechanisms that facilitate its invasion and prevent its recognition by host immune cells, i.e., using the low-level expression of viral antigens, the alternation of host cells’ gene expressions using perturbations of DNA methylation, and the downregulation of chemokines [14,16]. The downregulation of the host’s immune response by the HPV virus facilitates its persistence and further replication.

## 3. Primary Prevention of Cervical Cancer

The primary prevention of cervical cancer is achieved by avoiding HPV infection. The most effective method relies on HPV vaccination, especially in adolescents before their first sexual encounter. Currently, there are three vaccines approved by the U.S. Food and Drug Administration (FDA) that can be administered: bivalent (Cervarix), quadrivalent (Gardasil), and nonavalent (Gardasil 9) HPV vaccines. The first vaccines became available in 2006 and demonstrated a 90% efficiency rate in HPV infection prevention with HPV virus types 16 and 18 [17]. Additionally, both Gardasil vaccines protect against HPV infection genotypes 16 and 18 and 6 and 11, which cause 90% of genital warts. Gardasil 9 also protects against HPV genotypes 31, 33, 45, 52, and 58. A comparison of the vaccines is presented in Table 1.

All vaccines were developed using recombinant DNA technology. The vaccines were created from the purified self-assembling L1 protein, which forms HPV type-specific empty shells to mimic virus-like particles. As a result, the vaccines have high immunogenic potential and induce the organism production of HPV-specific antibodies, effectively preventing viral infection [18]. The vaccines are licensed to be administered in patients starting from the age of 9; however, they should be given following national guidelines and programs to ensure an optimal immunologic response while decreasing the likelihood of HPV infection [19].

The vaccines were found to reduce the risk of HPV infection, the development of HPV-related lesions, and the rates of relapse and subsequent HPV-related diseases after surgery for HPV-related diseases of the cervix and vulva. A post hoc analysis conducted by Joura et al. [20], based on randomized phase III controlled trials FUTURE I and FUTURE II, showed a significant effect of quadrivalent HPV vaccination and a reduction in the relapse of HSILs by 64.9%.

## 4. Secondary Prevention

The early detection and treatment of precancerous lesions are the methods of choice for cervical cancer prevention. The American Society of Colposcopy and Cervical Pathology (ASCCP) recently released the Risk-Based Management Consensus Guidelines for Abnormal Cervical Cancer Screening Tests and Cancer Precursors [21], which are the updated guidelines of the 2012 ASCCP management guidelines. The consensus provides recommendations for the management and treatment of patients with abnormal cervical cancer screening results.

The treatment of cervical dysplasia varies depending on the patient’s age, pregnancy status, the extent of cervical intraepithelial neoplasia, and the estimation of the patient’s risk of developing CIN 3+ based on a combination of medical results and history. In all nonpregnant patients, regardless of age, with a histopathological diagnosis of CIN 3, treatment is recommended, and observation is unacceptable (AII), with excisional treatment preferred to ablative treatment [21,22]. In nonpregnant patients with CIN 2, treatment is recommended unless the patient’s concerns concerning the influence of the treatment effect on future pregnancy outweigh concerns regarding cervical cancer (BII). For patients younger than 25 years old, observation with colposcopy, cytology, and HPV-based testing is possible [21].

Surgical excisional therapy can be performed using the loop electrosurgical excision procedure (LEEP), cold knife biopsy, or laser cone biopsy. Multiple studies have tried to determine the method of choice, but no technique has been found to be superior in terms of treatment failure or treatment-associated morbidity [23]. A meta-analysis conducted by Santesso et al. [24] demonstrated lower recurrence rates in patients who underwent cold knife conization than in those treated with LEEP or cryotherapy. However, the risk of CIN 2+ recurrence after surgical treatment remains at approximately 6.6% [25] and can be caused by residual disease due to the incomplete removal of the lesion, persistent infection in the surrounding tissues, reactivation of a latent HPV infection, or a new infection after the treatment with the same or other HPV types. Factors such as the patient’s age, size, and location; the severity of the intraepithelial lesion; the size of the excised specimen; complete lesion excision; surgical margin positivity; prior treatment and its modality; the presence of high-risk HPV after treatment; and the presence of comorbidities (autoimmune diseases, HIV, hepatitis B and/or C, malignancies, diabetes, genetic disorders, and/or history of organ transplant) were found to be significant independent predictors of residual/recurrent high-grade cervical intraepithelial neoplasia [25,26,27,28].

## 5. Secondary Vaccination

As HPV persistence is the leading and only modifiable factor affecting the risk of the progression of CIN lesions into high-grade cervical dysplasia and cancer, it has been proposed to conduct adjuvant vaccination in patients treated for high-grade cervical dysplasia. As no vaccine has been approved for therapeutic use in patients diagnosed with HSILs to date, attempts have been made to determine the use of HPV prophylactic vaccination in order to reduce recurrent HSILs and prevent cervical cancer. To date, numerous studies have been conducted, but only a few were prospective. In Table 2, we listed all up-to-date prospective studies regarding prophylactic HPV vaccination in patients diagnosed with CIN.

The mechanism of function of the HPV vaccine in patients already infected with HPV is not fully understood. The reduction in the recurrence rates of HPV infections and HPV-induced lesions may be due to the following: For patients who have not been previously vaccinated, the vaccination may act as primary prevention and protect against new HPV infections. Another mechanism may prevent the loss of immunological effectiveness against HPV reactivation/reinfection in patients who did not develop long-lasting immune protection after a previous infection [29].

Accumulating data show a potential role of HPV prophylactic vaccines in the adjuvant setting to surgical treatment in patients with HSILs; however, study results differ regarding vaccination efficacy. A recent meta-analysis by Di Donato et al. [36], conducted on 21,310 patients, revealed a significant risk reduction in recurrent CIN 1+ (OR 0.51, *p* = 0.006) and CIN 2+ (OR 0.35, *p* < 0.0001) after surgical treatment with adjuvant HPV vaccination compared with the unvaccinated group. The authors also noticed a nonsignificant reduction in HPV persistence among vaccinated patients. A different meta-analysis conducted by Jentschke et al. [37] showed a significant risk reduction in developing new HSILs after HPV vaccination independent of HPV type and patient age. Another recently published study by Kechagias et al. [38] also confirmed the reduction in the risk recurrence of CIN 2+ lesions in vaccinated patients compared to the unvaccinated population. The effect was found to be stronger when the risk recurrence was associated with HPV-disease-related subtypes of HPV16 and 18, but the confidence intervals were low, probably due to study inconsistency. Despite the encouraging data, it must be noted that many of the studies reported to date were either retrospective or post hoc analyses of studies, in which the study designs did not focus on adjuvant vaccine efficacy. Therefore, it is essential to evaluate and confirm the use of HPV vaccination as an adjuvant to surgical treatment in patients diagnosed with precancerous lesions divided into carefully selected groups in order to determine its perspective in everyday clinical use.

All up-to-date prospective studies have demonstrated a positive effect of adjuvant HPV vaccination and revealed lower rates of CIN 1+ and CIN 2+ recurrence. The most extensive study to date was a prospective cohort study conducted by Sand et al. [32], which included 17,128 patients who underwent conization for high-grade cervical dysplasia, of whom 2074 received HPV vaccination. The only retrospective study that did not report a significant risk reduction in CIN recurrence after surgical treatment and adjuvant vaccination was a study conducted by Hildesheim et al. [39]. However, this was a post hoc subgroup analysis of patients not separately randomized.

Post hoc studies of PATRICIA [40], FUTURE I, and FUTURE II clinical trials [20] analyzed the data concerning adjuvant HPV vaccination. However, the studies were conducted for different purposes, and the study designs excluded the initial enrolment of patients with a prior history of cervical lesions. In these trials, patients were vaccinated before conizations and were diagnosed with CIN lesions during the study duration. However, a positive effect of HPV vaccination was demonstrated, as patients undergoing treatment for cervical neoplasia after vaccination had a reduced risk of new or recurrent CIN 2+ development.

Similar studies regarding secondary HPV vaccination have been conducted in patients suffering from other HPV-related lesions. Adjuvant HPV vaccination after surgical treatment was found to reduce the incidence of subsequent vaginal intraepithelial neoplasia (VaIN), vulvar intraepithelial neoplasia (VIN), and genital warts [20]. Studies on the male population have also shown a decreased recurrence of genital warts after post-surgical HPV vaccination [41,42]. Other HPV-related cancers, including anal and laryngeal cancers, revealed similar benefits of adjuvant HPV vaccination for cancer prevention in patients diagnosed with precursor cancer lesions [43,44].

Even though there is no level I evidence for adjuvant HPV vaccination use in high-risk patients with HPV infection and HPV-associated lesions, multidisciplinary consensus evidence-based guidelines were created in Spain [45]. In accordance with them, HPV vaccination is strongly recommended in patients diagnosed with cervical precancerous lesions, and it is stated that the vaccine can be provided at any time but preferentially at diagnosis or before treatment.

## 6. HPV Status

The protective effect of secondary HPV vaccination seems to differ depending on HPV patient’s status. The most significant protective effect was demonstrated for patient-targeted HPV vaccine genotypes 16 and 18, giving a risk reduction of 63% [37]. However, all patients, independent of HPV type, benefited from adjuvant HPV vaccination after conization. Moreover, the first post-conization control at 6 months was shown to significantly impact the protective HPV vaccine effect. None of the patients who were demonstrated to have no disease at this time point (negative HPV test; negative pap test; and, when performed, a negative biopsy) were found to develop HSILs in the study follow-up period [33], supporting the hypothesis that the vaccination prevents the acquisition of new HPV infection. The study also demonstrated lower rates of persistent/recurrent HSILs in patients who had persistent LSILs/HSILs or HPV infection in the first post-conization control, but the results were not statistically significant. Studies conducted by Hogewoning et al. [46] and Munk et al. [47] have shown that consistent condom use increases the regression rates of CIN 2–3 lesions, most probably due to the reduction in the repetitive exposure of the cervical mucosa to an HPV-positive partner. Patient immunization against oncogenic HPV strains upon HPV vaccination, even if the patient has already been diagnosed with HSILs, should also result in a risk reduction in HPV transmission from HPV-positive sexual partners. 

## 7. Before or after Surgical Treatment?

Not only debatable is the use of the HPV vaccine in recurrent HPV, but also the timing of its administration. Persistent HPV infection seems to be associated with alternations in the local microenvironment and increased levels of proinflammatory cytokines [48,49]. The excision of HPV-related lesions causes modulation of the inflammatory environmental response and decreased cytokine levels. A study conducted by Saftlas et al. [50] demonstrated an immediate decrease in TNF-α in patients treated with LEEP to levels similar to those of untreated controls. As surgical treatment causes a change in the inflammatory tissue microenvironment, making it similar to that of HPV noninfected patients, it may be a good prerequisite for post-surgical vaccine intervention.

Different studies have used different vaccination timings. However, in most of them, the time of vaccine administration was either before LEEP/conization or shortly after (up to 1 month after LEEP). The study conducted by Sand et al. [32] revealed the possible implications of vaccination timing on its effect. Even though there was no statistically significant difference in the vaccination effect, the study showed a nonsignificant lower risk of recurrent HSILs in patients vaccinated 0–3 months before conization than in patients vaccinated 0–12 months after conization. Henere et al. [34] showed women vaccinated before surgical treatment (LEEP) to have a lower rate of post-treatment HSILs than non-vaccinated patients (0.9% vs. 6.5%, *p* = 0.047). The study evaluated 306 patients; however, patients with immunosuppression, multicentric HPV disease, a history of any HPV-associated cancer, and/or a diagnosis of invasive disease in the surgical specimen were excluded from the study. Moreover, the study did not specify the exact vaccination timing and only stated that “the first dose of the vaccine is provided immediately before or after the treatment according to the availability of the vaccine and the timing for HSIL treatment”. A meta-analysis revealed no significant effect of vaccination timing on the protective vaccination effect [37]; however, it did not include the recent study conducted by Henere et al. [34]. As HPV vaccination reduces the risk of future HPV infections and the future formation of new cervical lesions from newly occurring HPV infections, the vaccination should probably be administered as soon as possible. Further studies, especially randomized controlled trials, should be performed to determine the appropriate vaccination timing that would give optimal effects. Recent data have demonstrated that as many as 5–25% of patients post-conization may exhibit surgical margin positivity [51]. The incomplete excision of CIN lesions exposes patients to a substantial risk of post-treatment recurrent or persistent disease. A meta-analysis by Ghaem-Maghami et al. [52] revealed the importance of complete CIN excision, as high-grade post-treatment disease occurred in 18% of patients with a positive margin vs. 3% of females with complete excision. To date, there have been no studies comparing the effect of secondary vaccination in margin-positive and margin-negative patients. Moreover, none have evaluated its use adjuvant to reconization or any other form of treatment. In Table 3, we listed ongoing and upcoming clinical trials regarding the use of secondary HPV vaccinations.

## 8. Vaccination Valency and Dosage

In a study conducted by Del Pino et al. [33], vaccines of different valencies (2V, 4V, and 9V) were used, as the study was conducted in a real-life setting, and the vaccinations were not government-/study-founded. The study showed a similar protective effect of all HPV vaccine types on the risk of persistent and/or recurrent HSILs regardless of their valency. Moreover, no differences were observed between patients who received complete vaccination in a three-dose schedule and patients who obtained only one or two doses of the HPV vaccine [33]. The standard HPV vaccination protocol consists of a three-dose schedule at 0, 1–2, and 6 months. Some studies have demonstrated equivalent efficacies of the HPV vaccine in primary prophylactics when used either in a two-dose or one-dose schedule [53,54,55]. However, it must be noted that the studies were conducted on adolescent patients with no previous HPV infection, and the results may differ in a high-risk adult population and patients already diagnosed with cervical lesions.

## 9. Vaccination in HIV-Positive Patients

Immunocompromised patients, including patients with HIV, are more likely to have HPV infections and develop HPV-related lesions. In these patients, cervical HSILs are more likely to be caused by non-vaccine HPV types when compared to women without HIV [56]. The risk of developing HPV-associated cervical cancer in HIV-positive patients is 5-fold greater than in those without HIV infection [18], which is why this group of high-risk patients should be of particular importance. A meta-analysis by Debeaudrap et al. [57] showed an increased risk of residual or recurrent precancerous cervical lesion treatment failure in women with HIV (OR 2.7, 95% CI 2.0–3.5). A positive margin status was the only significant predictor of treatment failure, but HIV-positive patients tended to develop more extensive or multifocal lesions that can result in the incomplete removal of HSILs.

Firnhaber et al. [35] conducted the first prospective, randomized study evaluating the effect of HPV vaccination in HIV-positive patients diagnosed with HSILs. The authors found no significant difference in the prevention of recurrent HSILs after LEEP of HIV-positive patients between the control group and the group of patients in whom the quadrivalent vaccine was administered. In the study, 94% of patients were determined to be in HIV viral suppression and were treated with effective antiretroviral therapy (ART). The lack of effect of adjuvant vaccination may be related to an increased rate of patients with positive margins, which are risk factors for persistent and recurrent disease. Moreover, the vaccination timing in this study was different than in most of the trials on HIV-negative patients, as the first vaccination dose was administered before the surgical treatment of HSILs. This study is the only one that demonstrated no effect of HPV vaccination being administered to prevent recurrent HSILs after LEEP. However, it should be noted that only patients with HIV were included in the study. Further studies comparing the effect of HPV vaccinations in immunocompetent high-risk populations are required.

## 10. Vaccination Side Effects

Since the beginning of HPV vaccinations, more than 270 million doses of HPV vaccines have been distributed. Following the Global Advisory Committee on Vaccine Safety GACVS of the WHO, HPV vaccines were classified as extremely safe [58]. The safety profile was carefully established, as the vaccine is usually administered to adolescents or during potential childbearing years. The HPV vaccine side effects may include local reactions and may exert mild-to-moderate systemic effects based on the antigen quantity. The most common local adverse reactions are pain, swelling, and redness in the vaccinated area, while the systemic reactions include fever, nausea, fatigue, headache, and myalgia [59,60].

## 11. Therapeutic vs. Preventive Vaccines

Current HPV vaccines were engineered as primary preventive vaccines. Their function is to activate the patient’s humoral immunity and production of virus-neutralizing antibodies in order to prevent the viruses from entering host cells. The vaccines were found to be effective in protecting against persistent HPV infections and the formation of premalignant neoplasia lesions through the induction of neutralizing antibodies (IgG and IgA).

Therapeutic vaccines are constructed differently from prophylactic vaccines, as they should stimulate the cell-mediated immunity of the acquired immune cells, especially CD8+ T cells, rather than neutralizing antibodies [14]. They should aim to treat pre-existing HPV infections by stimulating dendritic cells and type T lymphocyte response against HPV antigens. Currently, there are no approved therapeutic vaccines against HPV, but multiple studies have investigated some possible candidate vaccines using different vaccine types and combination trials. Peptide-based, protein-based, viral vector, DNA virus-based viral vectors, RNA virus-based viral vectors, bacterial vectors, Listeria-based vectors, Lactobacillus-based vectors, cell-based, DNA-based, PHV DNA-based, HPV-DNA and immunogenic protein-based, RNA-based, and multi-platform vaccines have been proposed [61]. The majority of potential therapeutic vaccines concentrate on oncoproteins E6 and E7 as the target proteins responsible for the malignant transformation of HPV-related lesions [62]. To date, only two vaccines have been tested in phase III clinical trials on patients with CIN 1–3 lesions. MVA E2 is a cross-reactive E2 vaccine created using the vaccinia virus. It is the most tested vaccine, with 1356 patients being vaccinated as part of the phase III trial. The results are promising because, during the study, 89% of patients demonstrated complete elimination of intraepithelial lesions after treatment, and 81% of women cleared oncogenic HPV genotypes,; however, due to the lack of a control group, the actual efficacy could not be established [63]. Another candidate vaccine, VGX-3100, is a mixture of two plasmids containing codon-optimized sequences related to the E6 and E7 genes. The REVEAL 1 Study (NCT03185013), a phase 3 multi-center, randomized, double-blind, placebo-controlled trial, achieved primary and secondary efficacy endpoints of regression of cervical HSILs in 2021. The follow up of patients included in the study is still being continued simultaneously with the REVEAL 2 study to assess and confirm the vaccination’s safety, tolerability, and efficacy (NCT03721978).

The development of target vaccines requires additional time for the collection of confirmatory data and the creation of regulatory approval. In addition, despite the recent advanced techniques, the vaccines may not live up to the patient’s expectations due to their delivery methods, poor coverage of HPV genotypes, and limited safety data. Moreover, the success of the clinical trials may depend on patient selection criteria and the use of biomarkers of cancer invasiveness and prognosis to optimize patient selection and maximize potential therapeutic outcomes [64].

## 12. Conclusions

With the increasing evidence of a positive effect of secondary vaccination in patients with HPV-related lesions, there is a need for prospective studies evaluating the influence of prophylactic vaccines on relapsed and/or recurrent cervical intraepithelial neoplasia. Risk stratification and the selection of the response criteria of patients who would benefit the most are crucial for effective treatment and secondary prophylaxis. A better understanding of secondary vaccination, its timing, and its cost effectiveness are essential to developing international guidelines. Adjuvant HPV vaccination after surgical treatment may reduce the risk of recurrent cervical dysplasia.

## Figures and Tables

**Table 1 cancers-14-04352-t001:** A comparison of HPV vaccines.

	Valency	HPV Genotypes	Vaccination Schedule
Cervarix	Bivalent	16 and 18	0, 1, and 6 months
Gardasil	Quadrivalent	16, 18, 6, and 11	0, 2, and 6 months
Gardasil 9	Nonavalent	16, 18, 6, 11, 31, 33, 45, 52, and 58	0, 2, and 6 months

**Table 2 cancers-14-04352-t002:** Prospective studies evaluating the effectiveness of HPV vaccination after surgical treatment.

	Inclusion CRITERIA	Surgical Method	Study Design	Vaccination Type	Vaccination Timing	Study Population	Study Results
Ghelardi et al. [29]	CIN 2+ lesions/ stage IA1 cervical cancer	LEEP	Prospective case control SPERANZA study	Quadrivalent	30 days after LEEP, at 2 and 6 months after 1st dose	536 patients	Reduced risk of subsequent HSIL recurrence by 81.2% (95% CI, 34.3–95.7), irrespective of causal HPV type
Grześ et al. [30]	CIN I–CIN III, carcinoma in situ	LEEP, surgical conization	Prospective case control	Quadrivalent	-	75 patients	25 patients received vaccination; none had disease recurrence during the observation period
Pieralli et al. [31]	Patients treated for CIN with negative HPV test, cytology and colposcopy 3 months after treatment	Conization, other n.a.	Randomized controlled trial	Quadrivalent	3, 5, and 9 months after surgical treatment	178 patients	Disease recurrence rate significantly higher in non-vaccination group
Sand et al. [32]	CIN 2, CIN 3, carcinoma in situ	Conization	Prospective population-based cohort study	Not stated	0–3 months before or 0-12 months after conization	17,128 patients	Nonsignificant lower risk of CIN 2+ among vaccinated patients
Del Pino et al. [33]	CIN II-CINIII	Conization	Prospective	Bivalent, quadrivalent and nonavalent	Bivalent at 0, 1, and 6 months Quadrivalent at 0, 2, and 6 months Nonavalent at LEEP, and 2 and 6 months	265 patients	4.5-fold reduction in the risk of persistent/recurrent HSILs among vaccinated patients
Henere et al. [34]	HSIL	LEEP	Prospective	Nonavalent	Immediately before or after treatment, at 2 and 6 months	306 patients	Vaccination before treatment reduces the prevalence of post-treatment HSILs (2.6% vs. 10.5%)
Firnhaber et al. [35]	CIN 2-CIN 3 HIV positive	LEEP	Randomized, double-blind, placebo-controlled prospective clinical trial	Quadrivalent	1st dose 4 weeks before LEEP, week 4, and week 26	180 HIV-positive patients diagnosed with HSILs	No effect of HPV vaccination to prevent recurrent HSILs after LEEP in patients with HIV

**Table 3 cancers-14-04352-t003:** Registered clinical trials for recurrent or relapsed cervical intraepithelial neoplasia with prophylactic HPV vaccines.

Clinical Trial	Phase	Inclusion Criteria	Intervention	Recruitment Status	Estimated Study Start Date	Estimated Study Completion Date	Estimated Enrollment
NOVEL (NCT03979014)	Phase III	CIN 2–3 or AIS	GARDASIL 9 at the time of LEEP/ surgical conization	Not yet recruiting	1 November 2019	31 July 2023	1000 participants
COVENANT (NCT03284866)	Phase III	HIV-positive patients who present with HSILs	Gardasil 9 (at weeks 0, 4, and 26) + LEEP at week 4	Recruiting	31 July 2019	January 2024	536 participants
NCT02864147	Phase II	Patients with CIN 2-3 HPV+	Observation (control), imiquimod only, imiquimod + Gardasil 9	Recruiting	July 2016	January 2023	138 participants
HOPE9 (NCT03848039)	Phase III	HSILs or initially invasive cervical cancer (histological results ≥ CIN 2 + and ≤ Ia1	Gardasil 9 at months 0, 2 (day of LEEP), and 6	Not yet recruiting	December 2020	May 2028	1220 participants
VACCIN (Trial NL7938)	Phase III	CIN 2, CIN 3	Gardasil 9 at the time of LEEP, at 2 and 6 months	Recruiting	19 August 2019	August 2022	750 participants

Table in accordance with the available data as of 22 April 2022. AIS—adenocarcinoma in situ, HSIL—high-grade squamous intraepithelial lesion, CIN—cervical intraepithelial neoplasia.

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
