# Peer review of "Can Adjuvant HPV Vaccination Be Helpful in the Prevention of Persistent/Recurrent Cervical Dysplasia after Surgical Treatment?—A Literature Review"

_cancers, 2022, doi:10.3390/cancers14184352_

Round 1

Reviewer 1 Report

- In the introduction: Page 2 line 71, anal cancer should be also included;

Page 2 lines 73-75: Incorrect concept, invasive cancer is not developed due to continous metaplastic changes but to the accumulation of mutations

Page 3 line 88: Integration is not the only mechanism related to E6/E7 overexpression. The sentence should say ...The best known mechanism or most studied ...

Author Response

Dear reviewer, 

Thank you for your comments

We have improved the manuscript in accordance with your suggestions. We have also added some minor changes to the manuscript as proposed by the second reviewer.

Please see the improved version

Reviewer 2 Report

Michalczyk et al. have performed a literature review regarding HPV-vaccination after surgical treatment of high-grade cervical intraepithelial neoplasia (CIN2+). They have included seven prospective studies with a total of 18,668 patients evaluating the effectiveness of HPV vaccination after surgical treatment. They have also listet registered clinical trials for recurrent or relapsed cervical intraepithelial neoplasia with prophylactic HPV vaccines. In conclusion, with the increasing evidence of a positive effect of secondary vaccination in patients with HPV-related lesions, there is a need for prospective studies evaluating the influence of prophylactic vaccines on relapsed and/or recurrent cervical intraepithelial neoplasia.

Comments

1. There have been many published studies regarding HPV vaccination after conization, including reviews and meta-analysis. Michalczyk et al. have to include all relevant studies.

https://www.hpvworld.com/articles/hpv-vaccination-after-conization/

Jentschke M, Kampers J, Becker J, Sibbertsen P, Hillemanns P. Prophylactic HPV vaccination after conization: A systematic review and meta-analysis. Vaccine. 2020 Sep 22;38(41):6402-6409. doi: 10.1016/j.vaccine.2020.07.055. Epub 2020 Aug 4. PMID: 32762871.

https://pubmed.ncbi.nlm.nih.gov/32762871/

Kechagias K S, Kalliala I, Bowden S J, Athanasiou A, Paraskevaidi M, Paraskevaidis E et al. Role of human papillomavirus (HPV) vaccination on HPV infection and recurrence of HPV related disease after local surgical treatment: systematic review and meta-analysis BMJ 2022; 378 :e070135 doi:10.1136/bmj-2022-070135

https://www.contemporaryobgyn.net/view/hpv-vaccines-may-help-prevent-recurrence-of-high-grade-cervical-dysplasia

Casajuana-Pérez A, Ramírez-Mena M, Ruipérez-Pacheco E, Gil-Prados I, García-Santos J, Bellón-Del Amo M, Hernández-Aguado JJ, de la Fuente-Valero J, Zapardiel I, Coronado-Martín PJ. Effectiveness of Prophylactic Human Papillomavirus Vaccine in the Prevention of Recurrence in Women Conized for HSIL/CIN 2-3: The VENUS Study. Vaccines (Basel). 2022 Feb 14;10(2):288. doi: 10.3390/vaccines10020288. PMID: 35214747; PMCID: PMC8879017.

https://pubmed.ncbi.nlm.nih.gov/35214747/

Karimi-Zarchi M, Allahqoli L, Nehmati A, Kashi AM, Taghipour-Zahir S, Alkatout I. Can the prophylactic quadrivalent HPV vaccine be used as a therapeutic agent in women with CIN? A randomized trial. BMC Public Health. 2020 Feb 27;20(1):274. doi: 10.1186/s12889-020-8371-z. PMID: 32106837; PMCID: PMC7045378.

https://pubmed.ncbi.nlm.nih.gov/32106837/

2. In my opinion there is enough evidence to recommend HPV-vaccination to everybody regardless of HPV-status, with or without CIN, with or without treatment.

If the HPV-vaccine works to prevent future HPV-infections and future cervical lesions in women before sexual exposure, it will also prevent future HPV-infections and future cervical lesions from new HPV-infections in women with a positive HPV-test, in women with CIN and in women treated for CIN.

3. It should not be a question to vaccinate before or after surgical treatment. It is never a good idea to delay HPV vaccination. The vaccine should be given as soon as possible. During a lifetime, most women are exposed for HPV several times. The HPV-vaccine will reduce the risk of future HPV-infections and future cervical lesions from new HPV-infections.

4. The risk of exposure for HPV is not only related to new partner, but also in a stable relationship there will be transmission of HPV almost every time they have sexual intercourse as long as one of them have HPV. 

In a Norwegian study, women with CIN2/3 who were recommended use of condom, had a higher regression rate of CIN2/3 than women who not used a condom. The mechanism is probably reduced exposure of HPV. The HPV-vaccine will also reduce the risk of transmission from a partner. 

Munk AC, Gudlaugsson E, Malpica A, Fiane B, Løvslett KI, Kruse AJ, Øvestad IT, Voorhorst F, Janssen EA, Baak JP. Consistent condom use increases the regression rate of cervical intraepithelial neoplasia 2-3. PLoS One. 2012;7(9):e45114. doi: 10.1371/journal.pone.0045114. Epub 2012 Sep 13. PMID: 23028792; PMCID: PMC3441681.

https://pubmed.ncbi.nlm.nih.gov/23028792/

Minor revisions

Line 31, abstract, add "Adjuvant HPV vaccination after surgical treatment may reduce the risk of recurrent cervical dysplasia"

Line 61, introduction, "the recurrence or residual disease affects as much as 17% of patients" add the reference Katki 2013.

Katki HA, Schiffman M, Castle PE, Fetterman B, Poitras NE, Lorey T, Cheung LC, Raine-Bennett T, Gage JC, Kinney WK. Five-year risk of recurrence after treatment of CIN 2, CIN 3, or AIS: performance of HPV and Pap cotesting in posttreatment management. J Low Genit Tract Dis. 2013 Apr;17(5 Suppl 1):S78-84. doi: 10.1097/LGT.0b013e31828543c5. PMID: 23519309; PMCID: PMC3616418.

https://www.ncbi.nlm.nih.gov/pmc/articles/PMC3616418/

Line 66, introduction, "HPV is a family of more than 100 types" => "HPV is a family of more than 200 different HPV-types"

(Per August 2022 there is identified 229 different HPV-types)

https://www.hpvcenter.se/human_reference_clones/

Line 68-69, introduction, "The most common high-risk viruses responsible for approximately 70% of persistent HPV infections are HPV types 16 and 69 18, while types 31, 33, 45, 52, and 58 account for 19%" => "The two high-risk HPV-types responsible for approximately 70% of all cases of cervical cancer are HPV types 16 and 18, while types 31, 33, 45, 52, and 58 account for 19%"

Arbyn M, Tommasino M, Depuydt C, Dillner J. Are 20 human papillomavirus types causing cervical cancer? J Pathol. 2014 Dec;234(4):431-5. doi: 10.1002/path.4424. PMID: 25124771.

https://pubmed.ncbi.nlm.nih.gov/25124771/

Line 160, "the risk of CIN 2+ recurrence after surgical treatment remains at approximately 6.6%"

It is confusing to state that the risk is 6.6% when you in the introduction state 17%. I think you have to choose which reference you should use. In my experience the risk of recurrence is closer to 6.6% than 17%.

Bjørnerem MS, Sørbye SW, Skjeldestad FE. Recurrent disease after treatment for cervical intraepithelial neoplasia-The importance of a flawless definition of residual disease and length of follow-up. Eur J Obstet Gynecol Reprod Biol. 2020 May;248:44-49. doi: 10.1016/j.ejogrb.2020.03.022. Epub 2020 Mar 9. PMID: 32172024.

https://pubmed.ncbi.nlm.nih.gov/32172024/

Line 397-403, conclusions, include "Adjuvant HPV vaccination after surgical treatment may reduce the risk of recurrent cervical dysplasia"

Author Response

Dear reviewer, 

We would like to thank you for your comments

Ad 1. We agree with your comment, as multiple meta-analysis studies and systhematic reviews have already been published, yet there is a limited number of reviews discussing the role of HPV vaccination in adjuvant HSIL treatment - especially talking about different vaccination aspects including its valency, timing, use among high-risk population patients and so one. 

In our review, we discuss in depth only prospective studies, which are all listed in tables, but also other studies, included post-hoc analyses, meta-analyses, and systematic reviews are discussed throughout the manuscript.

The study by Jentsche, has already been cited and discussed in our review; please see line 202.

We added the meta-analysis by Kechagias et al. - it is a recently published study (as of August 2022) - the study included fewer population than the two previously mentioned meta-analysis by Di Donato and Jentshke and ended with similar conclusions - please see the paragraph we added lines 204-208

A study by Casajuana-Perez et al was a retrospective study. We decided to discuss further only prospective studies as we believe they are of the highest reliability. In Table 2 we list only prospective studies, which is mentioned in the Tables’ heading and lines 180-182. 

Other retrospective studies that were also published in the field include the studies by: Kang et al. (https://pubmed.ncbi.nlm.nih.gov/23623831/), Petrillo et al. (https://www.ncbi.nlm.nih.gov/pmc/articles/PMC7157656/), Przybylski et al. (https://journals.viamedica.pl/ginekologia_polska/article/view/GP.a2021.0164), Ortega et al., and Bogani et al.

The meta-analysis by Di Donato, Jentsche and Kechagias include the conclusions from both prospective, retrospective and post-hoc studies so the conclusions of all retrospective studies are also included in the study analysis. 

In our study, we did not include the study by Karimi-Zarchi et al. in our study as the study population was different - it did not include patients diagnosed with CIN lesions who underwent surgical treatment (LEEP/ surgical conization) and were vaccinated as a adjuvant method to the treatment despite its effect. The study by Karimi-Zachiri included patients “with histologically confirmed residual/recurrent CIN 1 or high grade CIN” and compared only the effect between the vaccinated and unvaccinated groups. Moreover the study did not talk about the methods of previous patient treatment and this is why we decided not to include this study into our review.

Ad 2. We also believe in the power of HPV vaccination. However, as we only review the literature in the field, we believe the specific recommendations and vaccination guidelines  should be formed by international groups of experts during specific panels i.e. of ASCO/ ESMO. 

Ad 3. Thank you for your comment. In paragraph 7, we discuss the available data in the field. We believe that in most of the centres the LEEP/ surgical conization treatment is made shortly form HSIL diagnosis and does not highly affect vaccination timing. However, due to the limited information in the field we would like to remain this paragraph. We added the following: “As HPV-vaccination reduces the risk of future HPV-infections and future formation of new cervical lesions from newly occurring HPV-infections, the vaccination should probably be administered as soon as available. “ - lines 290-292. 

Ad 4. Thank you for your comment. We added the following: “Studies by Hogewoning et al. and Munk et al. have shown consistent condom use to increase the regression rates of CIN 2-3 lesions, most probably due to the reduction of repetitive exposure of the cervical mucosa to HPV positive partner. Patient immunization against oncogenic HPV strains upon HPV-vaccination, even if the patient has already been diagnosed with HSIL lesion, should also result in risk reduction of HPV transmission from a HPV positive sexual partner.” - lines 262-267. Please see the improved version of the manuscript

Minor revisions

Line 31, abstract, add "Adjuvant HPV vaccination after surgical treatment may reduce the risk of recurrent cervical dysplasia” - we added the sentence

Line 61, introduction, "the recurrence or residual disease affects as much as 17% of patients" add the reference Katki 2013.

Katki HA, Schiffman M, Castle PE, Fetterman B, Poitras NE, Lorey T, Cheung LC, Raine-Bennett T, Gage JC, Kinney WK. Five-year risk of recurrence after treatment of CIN 2, CIN 3, or AIS: performance of HPV and Pap cotesting in posttreatment management. J Low Genit Tract Dis. 2013 Apr;17(5 Suppl 1):S78-84. doi: 10.1097/LGT.0b013e31828543c5. PMID: 23519309; PMCID: PMC3616418.

https://www.ncbi.nlm.nih.gov/pmc/articles/PMC3616418/ - due to your later comments we decided not to include this citation but to include the data suggested by Bjornerem et al as it includes newer data.

Line 66, introduction, "HPV is a family of more than 100 types" => "HPV is a family of more than 200 different HPV-types” - sorry for the mistake, we corrected the sentance

(Per August 2022 there is identified 229 different HPV-types)

https://www.hpvcenter.se/human_reference_clones/

Line 68-69, introduction, "The most common high-risk viruses responsible for approximately 70% of persistent HPV infections are HPV types 16 and 69 18, while types 31, 33, 45, 52, and 58 account for 19%" => "The two high-risk HPV-types responsible for approximately 70% of all cases of cervical cancer are HPV types 16 and 18, while types 31, 33, 45, 52, and 58 account for 19%"

Arbyn M, Tommasino M, Depuydt C, Dillner J. Are 20 human papillomavirus types causing cervical cancer? J Pathol. 2014 Dec;234(4):431-5. doi: 10.1002/path.4424. PMID: 25124771. - we have added this citation

Line 160, "the risk of CIN 2+ recurrence after surgical treatment remains at approximately 6.6%"

It is confusing to state that the risk is 6.6% when you in the introduction state 17%. I think you have to choose which reference you should use. In my experience the risk of recurrence is closer to 6.6% than 17%.

Bjørnerem MS, Sørbye SW, Skjeldestad FE. Recurrent disease after treatment for cervical intraepithelial neoplasia-The importance of a flawless definition of residual disease and length of follow-up. Eur J Obstet Gynecol Reprod Biol. 2020 May;248:44-49. doi: 10.1016/j.ejogrb.2020.03.022. Epub 2020 Mar 9. PMID: 32172024.

We decided to delete ref 6,7 and the one you suggested by Katki and decided to use the 6.6% with the reference Bjornerem et al. 

Line 397-403, conclusions, include "Adjuvant HPV vaccination after surgical treatment may reduce the risk of recurrent cervical dysplasia” - we added the sentence

Please see the improved version of the manuscript

Round 2

Reviewer 2 Report

All comments have been addressed. The response to the comments is adequate and improved the manuscript.

Minor revision

Line 12-13, simple summary, "the overall risk of recurrence of HSIL lesions remains at approximately 17%" => "the overall risk of recurrence of HSIL lesions remains at approximately 6.6%"

Line 360, vaccination side effects, "Since the begging of HPV vaccinations, more than 270 million doses of HPV vaccines have been distributed" => "Since the beginning of HPV vaccinations, more than 270 million doses of HPV vaccines have been distributed"

Author Response

Thank you for your comments, we corrected the manuscript. We have also corrected some additional minor spelling errors